# Human mtRF1 terminates COX1 translation and its ablation induces mitochondrial ribosome-associated quality control

Franziska Nadler[1], Elena Lavdovskaia[1,2], Angelique Krempler[1], Luis Daniel Cruz-Zaragoza[1], Sven Dennerlein[1] & Ricarda Richter-Dennerlein[1,2,3]✉

Translation termination requires release factors that read a STOP codon in the decoding center and subsequently facilitate the hydrolysis of the nascent peptide chain from the peptidyl tRNA within the ribosome. In human mitochondria eleven open reading frames terminate in the standard UAA or UAG STOP codon, which can be recognized by mtRF1a, the proposed major mitochondrial release factor. However, two transcripts encoding for COX1 and ND6 terminate in the non-conventional AGA or AGG codon, respectively. How translation termination is achieved in these two cases is not known. We address this long-standing open question by showing that the non-canonical release factor mtRF1 is a specialized release factor that triggers COX1 translation termination, while mtRF1a terminates the majority of other mitochondrial translation events including the non-canonical ND6. Loss of mtRF1 leads to isolated COX deficiency and activates the mitochondrial ribosome-associated quality control accompanied by the degradation of COX1 mRNA to prevent an overload of the ribosome rescue system. Taken together, these results establish the role of mtRF1 in mitochondrial translation, which had been a mystery for decades, and lead to a comprehensive picture of translation termination in human mitochondria.

Translation is a multistep high-fidelity process comprising initiation, elongation, termination and ribosome recycling. While tRNA adapter molecules undergo codon-anticodon interaction during initiation and elongation, specific proteinaceous factors, called release factors (RFs), recognize the STOP codon during translation termination in a sequence-dependent manner. RFs mimic tRNA-like structures that allow the interaction with the ribosomal A-site reaching the decoding center and the peptidyl transferase center (PTC). This results in a conformational change within the ribosome that mediates the hydrolysis of the ester bond between the nascent peptide chain and the peptidyl tRNA within the PTC. In bacteria there are two RFs: RF1 and RF2. While RF1 reads UAA and UAG codons, RF2 recognizes UAA and UGA. The codon specificity is defined by the codon-recognition motifs within domain 2, the PxT or SPF motif, respectively, and the tip of the α5 helix[1–4]. Both factors share the highly conserved GGQ motif within domain 3 that reaches into the PTC during termination and is essential for facilitating peptide hydrolysis. As mitochondria evolved from an α-proteobacterial ancestor, the human mitochondrial translation machinery reveals similarities to its bacterial counterpart, but also significant differences[5]. Human mitochondria use a slightly different genetic code, e.g. UGA encodes for tryptophan instead of being a STOP signal and AGA and AGG are not recognized by a tRNA$^{Arg}$, but terminate

[1]Department of Cellular Biochemistry, University Medical Center Goettingen, D-37073 Goettingen, Germany. [2]Cluster of Excellence "Multiscale Bioimaging: from Molecular Machines to Networks of Excitable Cells" (MBExC), University of Goettingen, D-37075 Goettingen, Germany. [3]Goettingen Center for Molecular Biosciences, University of Goettingen, D-37077 Goettingen, Germany. ✉e-mail: ricarda.richter@med.uni-goettingen.de

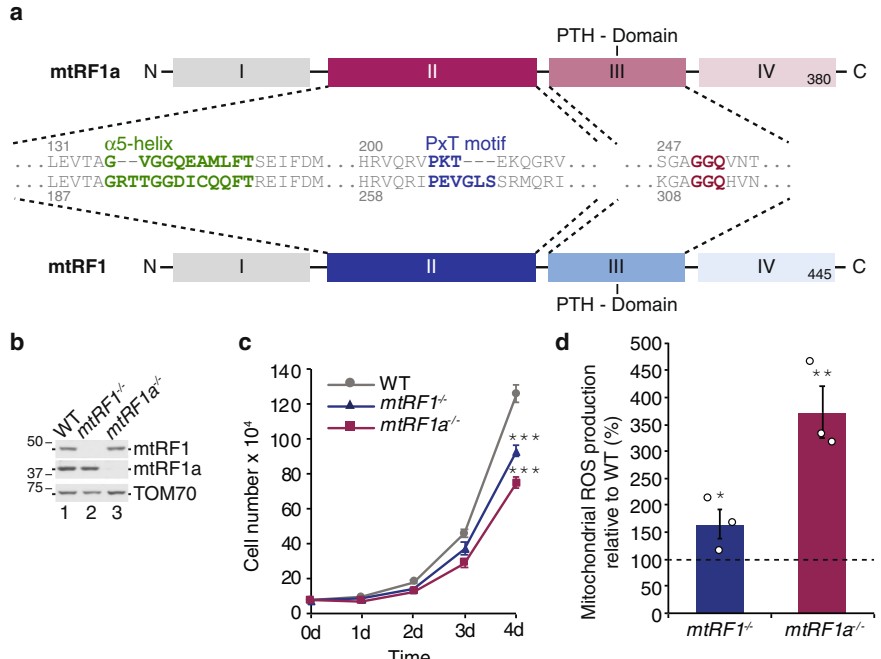

**Fig. 1 | Consequences of loss of mitochondrial translation termination factors.** **a** Comparison of mtRF1a (top, red colors) and mtRF1 (bottom, blue colors). The domains I–IV of the mitochondrial release factors are shown in boxes. Corresponding sequences of the decoding motifs in domain II (α5 helix: green; PxT motif: blue) and the GGQ motif in the peptidyl tRNA hydrolase (PTH) domain III are indicated. **b** CRISPR/Cas9-mediated knockout of mtRF1 and mtRF1a. Mitochondrial lysates (10 μg for mtRF1a and 75 μg for mtRF1) were separated by SDS-PAGE followed by western blotting and immunodetection using the indicated antibodies. **c** Ablation of mitochondrial release factors affects cellular growth. Cells ($7.5 \times 10^4$) were seeded on day 0. Cell growth was monitored by cell counts for the indicated time points in HEK293 wild-type cells (WT; gray) and in mtRF1- (*mtRF1⁻/⁻*, blue) and mtRF1a-deficient cells (*mtRF1a⁻/⁻*; red). ($n > 6$ biological replicates; mean ± SEM; significance for day 4 was calculated by two-sample one-tailed Student's *t*-test and defined as ***$p \leq 0.001$). **d** Elevated reactive oxygen species (ROS) production upon loss of mtRF1 and mtRF1a. ROS production was monitored by FACS using MitoSox Red. Relative ROS production in WT is indicated as dashed line (100%) and individual data points are shown as circles. Statistical analysis was carried out as two-sample one-tailed Student's *t*-test with $n = 3$ biologically independent samples and shown as mean ± SEM. Significance was defined as *$p \leq 0.05$; **$p \leq 0.01$. Source data are provided as a Source Data file.

the translation of *MTCO1* and *MTND6* transcripts (mRNA encoding COX1 and ND6)[6,7]. A proposed −1 ribosomal frameshift in both cases would allow the termination at a conventional UAG STOP codon[8]. Consequently, all thirteen mitochondrial DNA(mtDNA)-encoded proteins would terminate either in UAA or UAG. Four members of the release factor family have been identified in human mitochondria: mtRF1a, mtRF1, ICT1 (mL62) and C12ORF65 (mtRF-R)[9–12]. mL62 and C12ORF65 lack domains for codon specificity and are part of the ribosome rescue system in human mitochondria. While mL62 is a homolog of bacterial ArfB (alternative rescue factor B) recognizing stalled mitochondrial ribosomes with an empty A site[13], C12ORF65 is involved in the mitochondrial ribosome-associated quality control (mtRQC) acting on peptidyl tRNA moieties within split large mitochondrial ribosomal subunits (mtLSU)[14]. The overall domain architecture of human mtRF1a and mtRF1 is highly conserved and resembles the one of bacterial RF1[15]. However, mtRF1a reveals the highest sequence and structural similarity to bacterial RF1 (Supplementary Fig. 1) and indeed, terminates translation when recognizing UAA or UAG within the ribosomal A site[9,13]. The function of the fourth member mtRF1 remains a mystery since it was discovered in 1998[9,10]. It is an open question why do human mitochondria retain mtRF1, if mtRF1a is able to terminate all thirteen mtDNA-encoded peptides[5,8]. Compared to canonical RF1 and mtRF1a, mtRF1 shows insertions in the codon-recognition motifs: a PEVGLS hexapeptide instead of the PxT tripeptide motif and an insertion of two amino acids (RT) in the α5 helix (Fig. 1a). No release activity has been measured in vitro so far and also no particles of reconstituted 55 S mitochondrial ribosomes could be solved with bound mtRF1[9,13].

In this work, we determine the function of mtRF1 in human cells by generating a specific knockout cell line and subsequent investigation

of the consequences of loss of function in comparison to mtRF1a-ablated cells. Our results show that mtRF1 is required for mitochondrial function by ensuring proper COX1 translation termination. Although ablation of mtRF1 results in isolated COX deficiency, the activation of mtRQC prevents respiratory incompetence by partially rescuing stalled COX1-translating ribosome complexes.

## Results

### Human mtRF1 and mtRF1a are required for mitochondrial function

Although in vitro measurements and high-resolution structures reveal mechanistic insights into the function of mtRF1a, our knowledge regarding the cellular consequences of perturbed translation termination in human mitochondria is limited and only based on short-term siRNA-mediated depletion of mtRF1a[9]. To study the impact of loss of function of translation termination in human mitochondria, we have generated specific knockout cell lines using CRISPR/Cas9 technology. We used guide RNAs targeting exon 1 of mtRF1a and exon 2 of mtRF1, respectively; and isolated clones were confirmed by western blotting and genomic DNA sequencing (Fig. 1b, Supplementary Fig. 2a, b). In both cases premature STOP codons were identified leading to undetectable protein levels. Loss of both mitochondrial RFs affects cellular growth in high-glucose-containing media as the cell number over time is significantly reduced in comparison to the wild-type control, however, mtRF1a ablation is more severe than mtRF1 loss (Fig. 1c). Such growth defect has also been observed in other studies when interfering with mitochondrial translation[9,11,16–18]. In the absence of mtRF1a, cells tend to produce more reactive oxygen species (ROS), which is in agreement with previous observations and suggest mitochondrial dysfunction[9]. ROS production is less pronounced in *mtRF1⁻/⁻*, but still

significantly elevated compared to wildtype. Thus, both release factors are critical for mitochondrial function and cellular growth.

## Loss of human mtRF1 leads to isolated complex IV deficiency

In contrast to previous observations, mtRF1a-deficient cells are not able to respire as monitored by real time respirometry in intact cells (Fig. 2a). Therefore, these cells can only survive in the high-glucose containing media, in which they obtain their energy via glycolysis. *mtRF1a*$^{-/-}$ cells acidify the media relatively fast, indicating the conversion of excess pyruvate into lactate, a typical characteristic of oxidative phosphorylation (OXPHOS)-deficient cells. The oxygen consumption rate is also reduced in mtRF1-ablated cells in comparison to wildtype control, but does not reveal such a profound defect as *mtRF1a*$^{-/-}$ cells, which is in agreement with the relative growth rates. Nevertheless, these results show that both RFs are required for optimal OXPHOS capacity. To dissect how the loss of mtRF1a or mtRF1, respectively, affects the function of the different respiratory chain complexes, we monitored the amounts and the activity of complex I and IV by *in gel* activity measurements and in a colorimetric assay (Fig. 2b–d). While mtRF1a ablation results in an almost complete loss of complex I and IV activity, mtRF1 loss does not affect complex I, but reveals a significant reduction in complex IV activity. We further analyzed the individual complexes by Blue-Native PAGE and western blotting, confirming the combined OXPHOS deficiency in *mtRF1a*$^{-/-}$ with complex II, which is entirely nuclear (nDNA)-encoded, being unaffected (Fig. 2e). No respiratory supercomplexes are formed in the absence of mtRF1a and only the nuclear-encoded F$_1$ part of the ATP synthase is detectable providing the explanation for the drastic respiratory incompetence. In contrast, respiratory supercomplex formation is comparable between wildtype and *mtRF1*$^{-/-}$ cells. However, the dimeric complex IV at ~400 kDa is strongly reduced in mtRF1-deficient cells as visualized by COX1 antibody on BN PAGE (Fig. 2e, lane 11) and *in gel* activity indicating an isolated cytochrome *c* oxidase (COX) deficiency in *mtRF1*$^{-/-}$. Next, we investigated the protein steady-state levels of the mtDNA- and nDNA-encoded components of the OXPHOS complexes by western blotting (Fig. 2f, g). In agreement with reduced OXPHOS complexes, the tested mtDNA-encoded proteins are significantly reduced in *mtRF1a*$^{-/-}$ cells except COX1. In contrast, COX1 is strongly affected in *mtRF1*$^{-/-}$ while the other tested mtDNA-encoded proteins are unaltered. Similarly, nDNA-encoded structural OXPHOS components are significantly reduced in *mtRF1a*$^{-/-}$ while only marginally affected in *mtRF1*$^{-/-}$. We also investigated the steady-state levels of MITRAC (mitochondrial translation regulation assembly intermediate of cytochrome *c* oxidase) components, which form an assembly platform that coordinates COX1 synthesis with its subsequent assembly into complex IV. C12ORF62 and MITRAC12 interact with nascent COX1 ribosome complexes and represent essential assembly factors mediating the first steps during COX biogenesis[19–21]. Mutations in C12ORF62 or MITRAC12 lead to reduced COX1 synthesis and subsequently to isolated complex IV deficiency associated with severe neurological disorders in human patients[22,23]. While MITRAC12 is not drastically affected in either of the knockouts, C12ORF62 is altered. Whereas mtRF1 loss leads to reduced C12ORF62 levels, mtRF1a deficiency results in elevated amounts (Fig. 2h, i). We further elaborated these findings by 2D PAGE and reveal an accumulation of COX1-containing MITRAC complex at ~200 kDa in *mtRF1a*$^{-/-}$ while MITRAC is strongly reduced in *mtRF1*$^{-/-}$ (Fig. 2j). Thus, COX1 is trapped in MITRAC in mtRF1a-deficient cells, as further assembly steps are blocked due to reduced levels of other complex IV constituents such as COX2. In *mtRF1*$^{-/-}$ cells MITRAC is strongly reduced as indicated by decreased levels in COX1 and C12ORF62 suggesting defects in COX1 synthesis.

## mtRF1 is specifically required for COX1 translation

We measured the synthesis of mtDNA-encoded proteins by [$^{35}$S] Methionine de novo labeling and reveal that mtRF1a loss affects the translation of most mtDNA-encoded transcripts including the non-canonically terminated transcript of ND6, but not of COX1 (Fig. 3a). Contrary, the synthesis of COX1 is exclusively affected in *mtRF1*$^{-/-}$ while others are produced comparable to wild-type control (Fig. 3b). Nevertheless, COX1 is still produced in *mtRF1*$^{-/-}$ and is detectable in respiratory supercomplexes comparable to wild-type control (Fig. 2e) indicating a higher stability of the reduced newly synthesized COX1 in mtRF1-deficient cells. To prove this hypothesis we monitored the stability of COX1 by [$^{35}$S]Methionine pulse-chase experiment for 24 h. Indeed, COX1 reveals a higher stability in mtRF1-ablated cells compared to wild-type control (Supplementary Fig. 3a, b). These results suggest that the limiting amounts of COX1 are sequestered by respiratory supercomplexes to enhance its stability and to ensure respiratory competence in *mtRF1*$^{-/-}$. These findings are reminiscent to previous observations, where it has been reported that reduced levels of complex IV are preferentially assembled into respiratory supercomplexes also to ensure the assembly and stability of complex I[24–26].

Thus, both release factors are required for mitochondrial translation and the opposed effects indicate that the factors cannot compensate each other. To ensure that 55 S mitochondrial ribosomes are properly formed, we monitored ribosome particles by sucrose density ultracentrifugation (Fig. 3c). As 28 S small and 39 S large mitoribosomal subunits as well as 55 S ribosomes are detectable, defects in mitochondrial ribosome biogenesis are unlikely. To confirm that the translation defects are specific due to the loss of mtRF1a or mtRF1, respectively, and not caused by an off-target effect due to the CRISPR/Cas9 approach, we ectopically expressed the respective FLAG-tagged open reading frames in the knockout cell lines (Fig. 3d–g). In both cases the translation defect is rescued indicating that the knockouts are specific and the FLAG-tagged variants are functional.

Both mitochondrial RFs carry the highly conserved GGQ motif in domain 3 (Fig. 1a), which is essential to facilitate peptide hydrolysis within the ribosome in all kingdoms of life. Mutations within the GGQ motif in bacterial or eukaryotic release factors as well as in human mL62 disable RFs to terminate translation while the proteins are stably expressed and the interaction with the ribosome is maintained[11,27–30]. We expressed mitochondrial RF variants in which we mutated the two glycine residues of the GGQ motif into alanine residues (Fig. 3d–g). Mutant variants were expressed to comparable levels as the FLAG-tagged wild-type RFs. However, mitochondrial translation could not be restored indicating that the catalytic activity of both release factors is required for their function and for mtDNA-encoded protein synthesis. Thus, our data indicate that mitochondrial RFs are indeed required for mitochondrial translation termination in vivo and mtRF1 is specifically assigned for COX1 synthesis, while mtRF1a terminates other mitochondrial translation events including ND6.

## Ablation of mitochondrial release factors affect mt-mRNA stability

Defects in mitochondrial translation often lead to an upregulation of mitochondrial transcripts potentially as a compensatory effect as indicated in previous studies[31,32]. We also investigated the steady-state levels of mitochondrial mRNAs as well as rRNAs by northern blotting in *mtRF1*$^{-/-}$ and *mtRF1a*$^{-/-}$ (Fig. 4a, b). The 12 S (*MTRNR1*) and 16 S rRNA (*MTRNR2*) remain stable in both knockouts, which is in agreement with the proper formation of mitochondrial ribosomes (Fig. 3c). However, mt-mRNAs are contrarily affected upon loss of mitochondrial RFs. While COX1-encoding mRNA (*MTCO1*) is significantly reduced in mtRF1-ablated cells, it remains stable in *mtRF1a*$^{-/-}$. In contrast, transcripts encoding for COX2 or CYTB are strongly decreased in *mtRF1a*$^{-/-}$, but are unaffected upon mtRF1 loss. Mitochondrial transcripts derive from two polycistronic transcripts. With the exception of *MTND6* (mRNA encoding for ND6), all of the mt-mRNAs arise from the polycistronic transcript synthesized from the heavy strand. If the loss of mitochondrial RFs would affect mitochondrial transcription, one

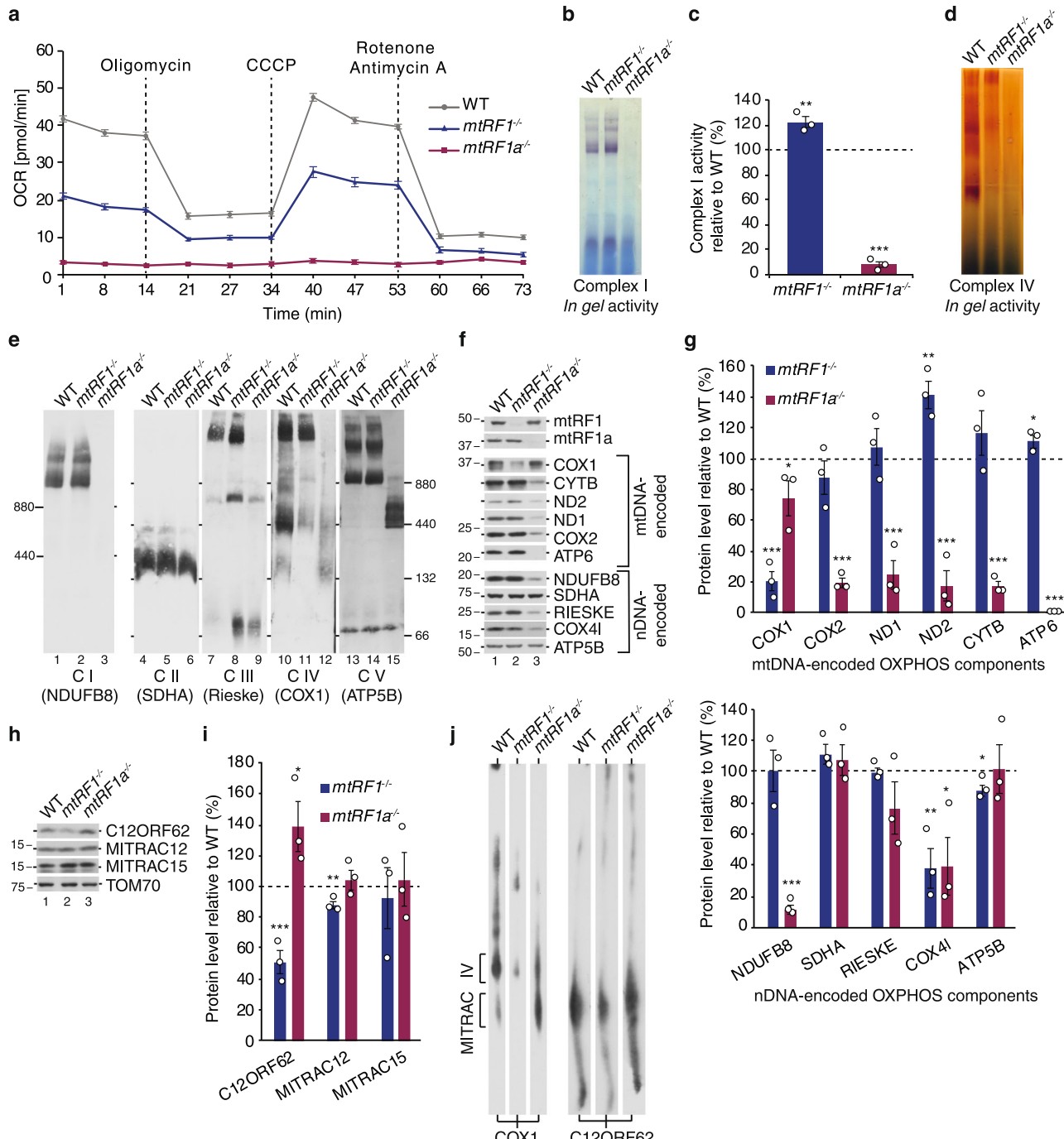

**Fig. 2 | Ablation of mitochondrial release factors affects OXPHOS. a** Oxygen consumption rate (OCR) is altered in *mtRF1*−/− and *mtRF1a*−/−. OCR was measured for the indicated time in wild-type cells (WT, gray), *mtRF1*−/− (blue) and *mtRF1a*−/− (red). Complex V was blocked using oligomycin, the membrane potential was uncoupled using CCCP and the activity of complex I and III was diminished by the addition of rotenone and antimycin A, respectively. **b−d** The activity of complex I and IV was measured by *in gel* activity (**b, d**) or in a colorimetric assay (**c**). **c** Complex I activity of WT is indicated as dashed line and individual data points are shown. Statistical analysis was performed as two-sample one-tailed Student's *t*-test with *n* = 3 biologically independent samples shown as mean ± SEM. Significance was defined as **$p ≤ 0.01$; ***$p ≤ 0.001$. **e** The effect of mtRF1 and mtRF1a loss on OXPHOS complexes. Isolated mitochondria (30 μg) were separated by BN-PAGE (2.5−10% gradient gel: CI; 4−14% gradient gel: CII-CV) followed by western blotting and

immunodetection using the indicated antibodies in brackets for complex I−V. **f−i** Loss of release factors affects mtDNA-encoded OXPHOS components. Mitochondria were analyzed by western blotting using the indicated antibodies for mtDNA- and nDNA-encoded OXPHOS components (**f, g**) as well as for MITRAC constituents (**h, i**). The relative protein steady-state levels were measured from three independent experiments and are presented as mean ± SEM (individual data points are shown as circles) relative to WT indicated as dashed line (100%). Statistical analysis was performed as two-sample one-tailed Student's *t*-test and significance was defined as *$p ≤ 0.05$; **$p ≤ 0.01$; ***$p ≤ 0.001$. **j** MITRAC is contrarily affected in mtRF1- and mtRF1a-ablated cells. Isolated mitochondria (150 μg) were subjected to BN-PAGE in the first dimension followed by SDS-PAGE in the second dimension prior to immunoblotting. Source data are provided as a Source Data file.

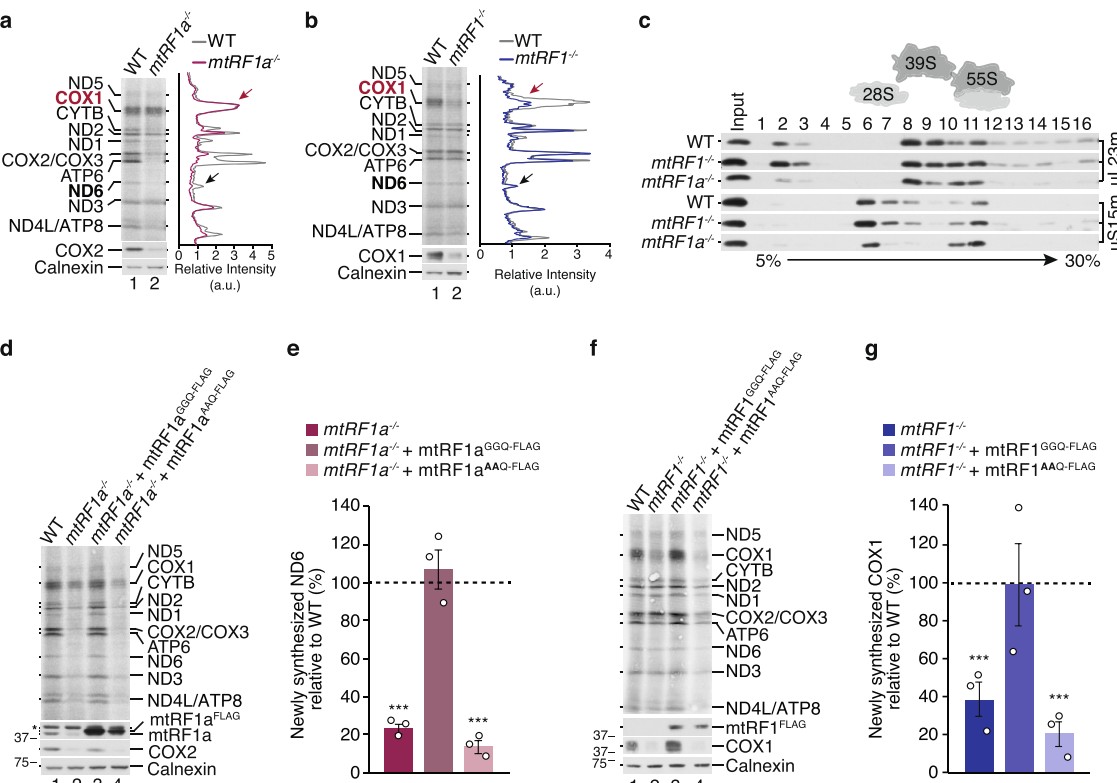

**Fig. 3 | Mitochondrial translation requires mtRF1 and mtRF1a. a, b** The synthesis of mtDNA-encoded proteins was analyzed by [³⁵S]Methionine de novo incorporation in wild-type (WT), *mtRF1a⁻/⁻* (**a**) and *mtRF1⁻/⁻* cells (**b**). Samples were analyzed by autoradiography and western blotting, respectively, and relative intensity was measured using ImageJ software. Similar results were obtained in n ≥ 3 biologically independent experiments. **c** Loss of mitochondrial release factors does not affect mitochondrial ribosome biogenesis. Purified mitoplasts (500 µg) from indicated cell lines were separated by sucrose density ultracentrifugation. Collected fractions (1–16) were analyzed by western blotting using uL23m as a marker of the large mitoribosomal subunit (mtLSU) and uS15m to indicate the small mitoribosomal

subunit (mtSSU). Input = 10% of total. Similar results were obtained in n ≥ 3 biologically independent experiments. **d–g** The GGQ motif is required for mtRF1 and mtRF1a function. C-terminal FLAG-tagged variants of mtRF1a and mtRF1 were ectopically expressed in the respective knockout cell line. Mitochondrial translation was monitored as in **a**. Newly synthesized ND6 (**e**) and COX1 (**g**) were measured in the indicated cell lines (individual data points are shown as circles) and calculated relative to WT (dashed line, 100%). Statistical analysis was performed as two-sample one-tailed Student's *t*-test with n = 3 biologically independent samples and shown as mean ± SEM. Significance was defined as ***p ≤ 0.001. Asterisk in (**d**) indicates unspecific signals. Source data are provided as a Source Data file.

would expect an overall decrease in all mitochondrial transcripts. However, as we observe a selective decrease in specific transcripts in the individual knockouts, we conclude that it is more likely an issue of RNA stability rather than synthesis. To support this hypothesis, we blocked mitochondrial translation using chloramphenicol and show that *MTCO1* levels can be restored in *mtRF1⁻/⁻* (Fig. 4c, d) indicating that the degradation of *MTCO1* transcripts is dependent on translation. To ensure that *MTCO1* is the only affected transcript in *mtRF1⁻/⁻*, we measured the steady-state level of all mt-mRNAs by NanoString analysis (Fig. 4e). In agreement with our northern blot results, *MTCO1* was the only reduced transcript in mtRF1-ablated cells whereas all the other transcripts were comparable to wildtype. Thus, mtRF1 loss induces the degradation of *MTCO1*, which might be part of a quality control system. Nevertheless, COX1 is not completely diminished and still detectable in mitochondrial supercomplexes in the absence of mtRF1, which ensures respiratory competence in *mtRF1⁻/⁻*. Consequently, an alternative factor must fulfill the task of COX1 translation termination and thus compensate for the loss of mtRF1.

### Loss of mtRF1 induces mitochondrial ribosome-associated quality control

A potential alternative factor responsible for the release of newly synthesized COX1 if mtRF1 is missing might be another member of the mitochondrial release factor family. We measured the steady-state level of mtRF1a, mL62 and C12ORF65 in *mtRF1⁻/⁻* and reveal significant elevated levels of C12ORF65 in mtRF1-ablated cells suggesting that the

mitochondrial ribosome-associated quality control machinery is responsible for the rescue of stalled COX1-translating ribosomes (Fig. 5a, b). Levels of mL62, which represents another system to rescue ribosomes stalled on truncated mRNAs, are comparable to wildtype. As mL62 requires an empty A site, it seems to be less likely that mL62 would rescue COX1-translating ribosomes in *mtRF1⁻/⁻*. mtRF1a also appears unaltered in *mtRF1⁻/⁻* and our results actually suggest that mtRF1a and mtRF1 cannot compensate each other. Therefore, C12ORF65, which together with MTRES1 represents the mtRQC, is a promising candidate to facilitate the release of COX1 from the ribosome if mtRF1 is missing. We tested this hypothesis by depleting C12ORF65 in *mtRF1⁻/⁻* and monitored the level of newly synthesized COX1 upon [³⁵S]Methionine metabolic labeling (Fig. 5c, d). C12ORF65 is efficiently downregulated and also leads to a decrease in MTRES1 indicating an interdependence of these factors. In line with our assumption, COX1 synthesis is significantly more decreased upon C12ORF65 depletion in *mtRF1⁻/⁻* than in non-targeting siRNA-treated knockout cells. Thus, loss of mtRF1 induces mtRQC to compensate for deficient termination events during COX1 translation in *mtRF1⁻/⁻*.

## Discussion

The role of mtRF1 during translation termination in human mitochondria was a long standing open question since it was discovered in 1998[10]. Here, we show that mtRF1 is required for COX1 translation termination and thus for cytochrome *c* oxidase function (Fig. 6). MITRAC is the first assembly platform for newly synthesized COX1, where nascent COX1

interacts with the early MITRAC constituents C12ORF62 and MITRAC12[19,20]. Both factors are associated with severe mitochondrial diseases with isolated COX deficiency[22,23]. While patients with mutation

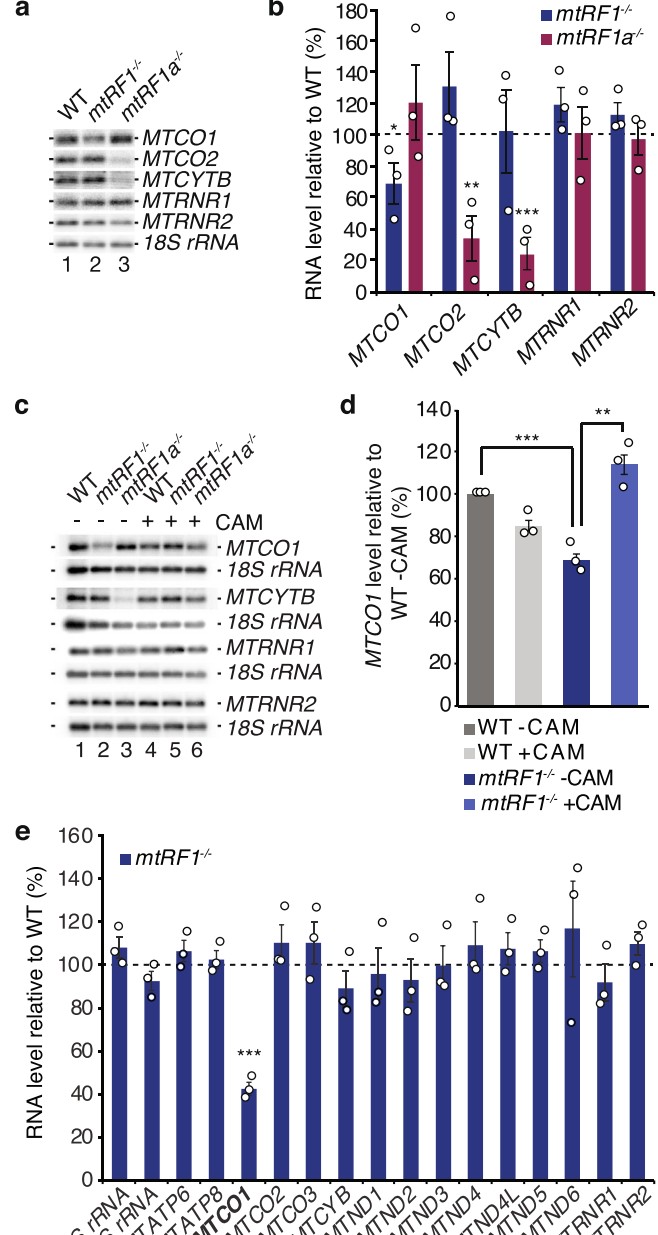

**Fig. 4 | Mitochondrial transcripts are altered upon ablation of mtRF1 or mtRF1a. a, b** RNA was isolated from wild-type (WT) cells, *mtRF1⁻/⁻* and *mtRF1a⁻/⁻* and subjected to northern blot analysis. Mitochondrial mRNAs encoding for COX1 (*MTCO1*), COX2 (*MTCO2*) and CYTB (*MTCYTB*) as well as the mt-rRNAs 12 S rRNA (*MTRNR1*) and 16 S rRNA (*MTRNR2*) were detected using specific probes. 18 S rRNA was used as loading control. RNA levels are calculated relative to WT (dashed line) and individual data points are indicated as circles. Statistical analysis was performed as two-sample one-tailed Student's *t*-test with $n = 3$ biologically independent samples and shown as mean ± SEM. Significance was defined as *$p \leq 0.05$ ; **$p \leq 0.01$ ; ***$p \leq 0.001$. **c, d** RNA was isolated from WT, *mtRF1⁻/⁻* and *mtRF1a⁻/⁻* treated with chloramphenicol (CAM, 50 μg/ml) for 24 h as indicated. Northern blot and statistical analysis were performed as in **a, b. e** Isolated RNA from WT and *mtRF1⁻/⁻* mitochondria was subjected to NanoString analysis. 18 S and 5 S rRNA were used as controls. The relative RNA levels of WT are indicated as dashed line. Statistical analysis was performed as two-sample one-tailed Student's *t*-test with $n = 3$ biologically independent samples and shown as mean ± SEM. Significance was defined as ***$p \leq 0.001$. Source data are provided as a Source Data file.

in C12ORF62 display fatal neonatal lactic acidosis, mutations in MITRAC12 are associated with neuropathy and exercise intolerance. As the loss of mtRF1 leads specifically to a reduction in COX1 and subsequently to decreased levels of C12ORF62, mtRF1 is a potential candidate when screening patients with isolated COX deficiency.

Our data also show the physiological importance of mtRQC during mitochondrial translation termination. It has been recently demonstrated that C12ORF65 is part of the mtRQC and required as a rescue factor for stalled ribosome complexes under conditions of aminoacyl tRNA starvation[14]. This population of stalled ribosomes with intact mRNA are not a preferred substrate for mL62, which requires an empty A site to protrude its C-terminal tail into the mRNA channel, similarly to its bacterial counterpart ArfB[13,33]. This makes it unlikely that mL62 rescues COX1-translating ribosomes in *mtRF1⁻/⁻* as the A-site would be still occupied with mRNA. The mtRQC would first allow the dissociation of these stalled complexes into the large (mtLSU) and the small mitochondrial ribosomal subunit (mtSSU) followed by the binding of C12ORF65 and MTRES1 to the split mtLSU with the peptidyl tRNA in the P-site (Fig. 6). Finally, C12ORF65 would facilitate the hydrolysis of nascent COX1 from the tRNA. Thus, the activation of mtRQC in *mtRF1⁻/⁻* partially compensate for the abolished COX1 translation termination by mtRF1. This enables mitochondria to produce a certain fraction of COX1, which assembles as part of complex IV into the stable supercomplexes capable of respiration, although to a reduced level compared to wildtype. In agreement with previous studies is that the reduced COX1 tends to be assembled within the supercomplexes and not within the free complex IV, which likely enhances the stability of the reduced newly synthesized COX1 in mtRF1-deficient cells[24–26]. The reduction of mtRQC by siRNA-mediated depletion of C12ORF65 in *mtRF1⁻/⁻* shows further reduction in COX1 translation indicating the importance of mtRQC for mitochondrial function. This central importance of mtRQC is also demonstrated by a growing group of patients with mutations in *C12orf65* developing Leigh syndrome[12,34]. It is currently unknown which factor acts upstream of C12ORF65-MTRES1, allowing the dissociation of the ribosome into the subunits. Besides the canonical recycling system composed of mtRRF and mtEFG2, the alternative recycling factor GTPBP6 is a potential candidate to be part of the mitochondrial ribosome rescue system[5,17,35]. However, both recycling pathways do not prefer ribosomes with a peptidyl tRNA in the P-site. Thus, a so far unidentified factor might be part of the mtRQC, facilitating the dissociation of the ribosome prior to binding of C12ORF65 to the mtLSU.

The decrease in *MTCO1* suggests a feedback mechanism that allows the specific degradation of this transcript in *mtRF1⁻/⁻*, potentially to prevent an overload of the mtRQC (Fig. 6), which already responds with a higher expression of C12ORF65 to cope with these stalling events. Consequently, mtRQC seems to involve not only the rescue of stalled ribosomes but also the degradation of the respective mRNA to avoid mtRQC stress and to minimize proteotoxic burden. A similar scenario would apply if mtRF1a is missing and e.g. *MTCO2* is degraded to avoid too many stalling events of COX2-translating ribosomes. Contrarily to *mtRF1⁻/⁻*, COX1 remains stable in *mtRF1a⁻/⁻*, but becomes stalled in the MITRAC complex as COX2 and COX3 translation are diminished (Fig. 6). Thus, both mitochondrial RFs exhibit substrate specificity and cannot compensate each other.

COX1 is a rather unusual case as it terminates in AGA (Supplementary Fig. 4)[6]. However, a −1 ribosomal frameshift would allow the termination in the conventional STOP codon UAG[8]. It is still under debate whether a −1 ribosomal frameshift is really occurring during COX1 and also during ND6 translation termination or whether a specialized factor might be responsible in reading AGA and AGG codons. In the past it has been suggested that mtRF1 might be able to recognize these unconventional STOP signals[9], although experimental evidence is completely missing. Nevertheless, bioinformatic studies using homology modeling and molecular dynamics simulation suggest that

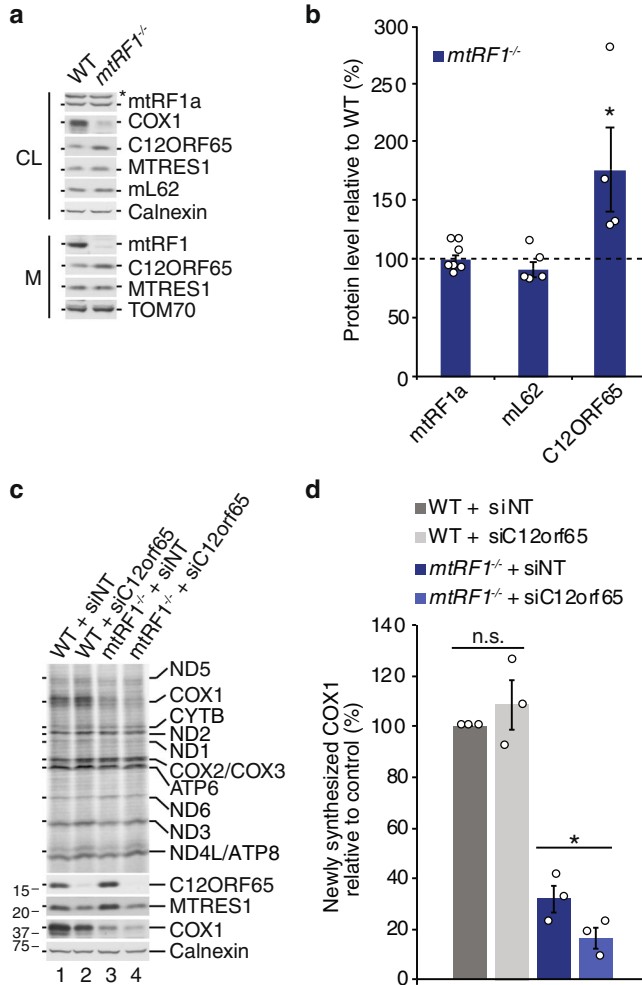

**Fig. 5 | Loss of mtRF1 activates C12ORF65. a, b** C12ORF65 protein levels are elevated in mtRF1-deficient cells. Whole cell lysates (CL, 25 µg or 50 µg) or isolated mitochondria (M, 75 µg) were analyzed by western blotting using the indicated antibodies. Protein levels were quantified and calculated relative to the wild-type control (WT; dashed line; 100%) and individual data points are shown as circles. Statistical analysis was performed as two-sample one-tailed Student's *t*-test with $n \geq 4$ biologically independent samples shown as mean ± SEM. Significance was defined as *$p \leq 0.05$. Asterisk in **a** indicates unspecific signals. **c, d** Depletion of C12ORF65 in *mtRF1*$^{-/-}$ further reduces COX1 levels. WT and *mtRF1*$^{-/-}$ cells were transfected with siNT or siC12orf65. After 3 days mitochondrial protein synthesis was monitored as described in Fig. 3 and efficiency of siRNA-mediated depletion was controlled by western blotting. Newly synthesized COX1 was quantified from three independent experiments shown as mean ± SEM with individual data points indicated as circles. Statistical analysis was performed as two-sample one-tailed Student's *t*-test and significance was defined as *$p \leq 0.05$ and $p > 0.05$ as not significant (n.s.). Source data are provided as a Source Data file.

the non-canonical mtRF1 with its insertion in the α5 helix and the PEVGLS motif co-evolutionary adapted to the changes within the mitoribosome and that it has a preference for UAA and UAG codons and neither for AGA or AGG or an empty A site[15]. This actually supports the −1 ribosomal frameshift hypothesis. Additionally, COX1 terminates in UAA and not in AGA in other species including mice, rat or bovine, although they also possess mtRF1 and mtRF1a (Supplementary Fig. 4, 5). Thus, if mtRF1 is a specialized release factor for COX1 termination in other species as well and if one considers the bioinformatic preference for UAA and UAG STOP codon by mtRF1, then this would be in favor with the −1 ribosomal frameshift theory in human mitochondria that allows termination in UAG. Along this line, we observed a decrease in ND6 in *mtRF1a*$^{-/-}$, but no in *mtRF1*$^{-/-}$ suggesting that mtRF1a is

responsible for ND6 termination (Fig. 3a, b). However, mtRF1a reads specifically UAA and UAG codons, but does not exhibit release activity on AGG codon[9,11], which also supports the hypothesis that a −1 ribosomal frameshift allows the conventional termination of ND6 in UAG recognized by mtRF1a.

Taken together, we have solved the mystery of the function of mtRF1 in human mitochondria and show that mtRF1 is specifically responsible for the termination of COX1 translation, which likely requires a −1 ribosomal frameshift by the human mitochondrial ribosome.

## Methods

### Key reagents
An extended table of plasmids, oligonucleotides, antibodies and other materials used is provided in Supplementary Table 1 and 2.

### Cell culture
HEK293 (Human Embryonic Kidney 293-Flp-In T-Rex, Thermo Fisher Scientific) cells were cultured in DMEM (Dulbecco's modified Eagle's medium) supplemented with 10% [v/v] FCS (Fetal Calf Serum), 2 mM L-glutamine, 1 mM sodium pyruvate and 50 µg/ml uridine at 37 °C under 5% CO$_2$ humidified atmosphere. Cells were regularly monitored for the absence of Mycoplasma by GATC Biotech.

HEK293 mtRF1 and mtRF1a knockout cell lines were generated using Alt-R CRISPR/Cas9 technology (Integrated DNA Technologies, IDT) according to the manufacturer's instructions. In brief, cells were co-transfected with crRNA-tracrRNA duplex and Cas9 nuclease. The crRNA was designed to target the first or second exon of either the mtRF1a or mtRF1 gene, respectively. Clones were screened by immunoblotting and verified by TOPO cloning and subsequent sequencing. Usage of the TOPO™-TA Cloning™ Kit (Thermo Fisher Scientific) allows simple analysis of gDNA from respective clones. TOPO cloning is based on ligation of the amplified PCR product of the gDNA sequence targeted by the CRISPR guide RNA into a pCR4-TOPO TA vector. After transformation into OneShot™ competent *E.coli* cells, clones were selected on ampicilin LB-Agar plates. Picking a statistical relevant number of clones (≥20), their plasmid DNA were sequenced using M13 forward and reverse primers, allowing analysis of occurring INDELs in the genome caused by CRISPR/Cas9 technology.

Stable inducible cell lines expressing C-terminal FLAG-tagged versions of mtRF1 or mtRF1a were generated following established protocols[19,20]. Briefly, HEK293 cell lines were transfected with pOG44 and pcDNA5/FRT/TO plasmids harboring respective FLAG constructs using Lipofectamine3000 (Invitrogen) as transfection reagent according to the manufacturer's instructions. Cells were selected using 100 µg/ml hygromycin B.

Transient siRNA-mediated knockdown was performed by transfecting HEK293 cells with 33 nM siRNA oligonucleotides (Eurogentec) targeting the transcript of interest (see Supplementary Table 1) or non-targeting siRNA as control by using Lipofectamine RNAiMax (Invitrogen) as transfection reagent. Cells were incubated for 72 h at 37 °C under 5% CO$_2$ humidified atmosphere prior to further investigation.

### Measurements of mitochondrial radicals
Cells ($10^6$) were stained with 5 µM MitoSox Red (Invitrogen) according to the manufacturer's instructions to detect ROS. For flow cytometry analysis, 10,000 gated events were recorded on a BD FACS Canto II (Becton Dickinson) and analyzed using FACS-Diva software.

### Respirometry
Oxygen consumption rates (OCR) were measured using a XF96 Extracellular Flux Analyzer (Seahorse Bioscience). Cells ($4 \times 10^4$ per well) were directly seeded in assay buffer into a 96-well sample plate, spun down and incubated for 1 h in a non-CO$_2$ incubator at 37 °C

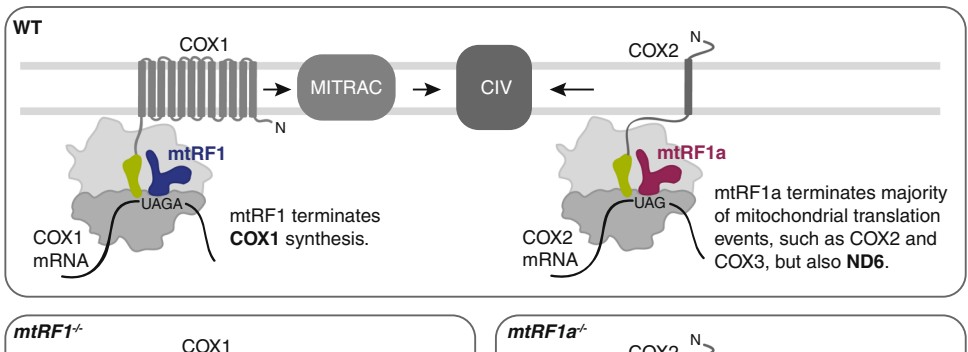

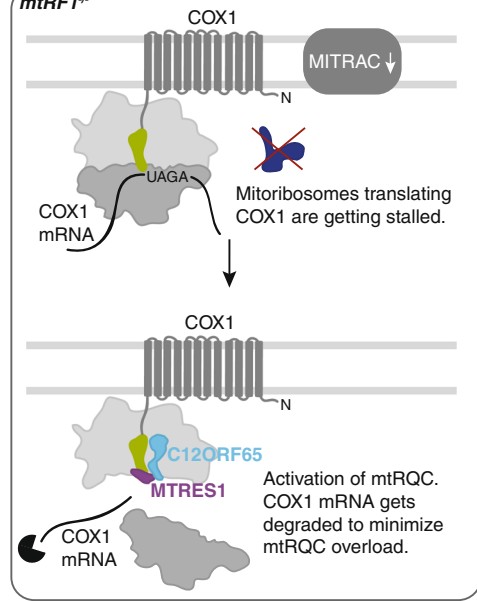

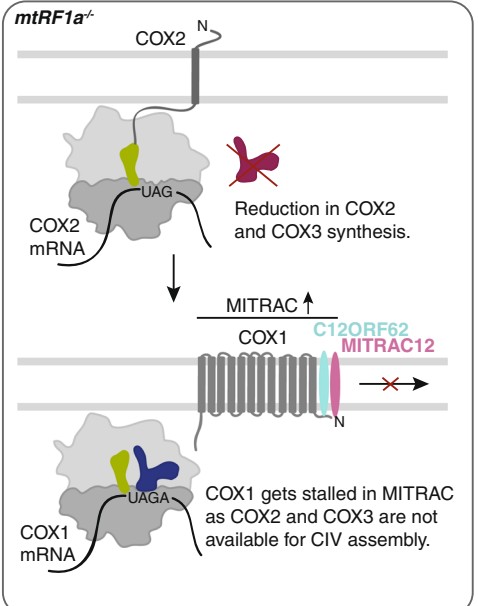

**Fig. 6 | The role of mtRF1 and mtRF1a during translation termination in human mitochondria.** mtRF1 (blue) specifically terminates COX1, whose synthesis and subsequent assembly is coordinated by MITRAC. In contrast mtRF1a (red) facilitates the release of COX2 and other mtDNA-encoded proteins including ND6. In the absence of mtRF1, termination of COX1 translation is affected and ribosomes are getting stalled. The decrease of newly synthesized COX1 leads to a reduction of MITRAC in *mtRF1*[−/−]. Stalled COX1-translating ribosomes are rescued by the mitochondrial ribosome-associated quality control (mtRQC) composed of C12ORF65 (light blue) and MTRES1 (purple). COX1 mRNA gets degraded in *mtRF1*[−/−] to reduce stalling events and therefore to prevent a mtRQC overload. Loss of mtRF1a does not affect COX1 translation termination, but causes severe reduction in COX2 and COX3. Thus, synthesized COX1 gets stalled in MITRAC as COX2 and COX3 are not available for further complex IV assembly in *mtRF1a*[−/−].

before basal respiration was measured. Subsequent automated addition of 3 μM oligomycin, 1.5 μM CCCP and 1 μM antimycin A plus 1 μM rotenone was used to monitor maximal respiration.

### Preparation of whole cell lysates, isolation of mitochondria from cultured cells and immunodetection via western blotting

Lysis of whole cells was carried out in nonionic lysis buffer (50 mM Tris-HCl pH 7.4, 130 mM NaCl, 2 mM MgCl₂, 1% NP-40, 1 mM PMSF and 1x Protease Inhibitor (PI) Cocktail (Roche)). For isolation of mitochondria, cultured cells were harvested and resuspended in homogenization buffer (300 mM trehalose, 10 mM KCl, 10 mM HEPES-KOH pH 7.4) with 1 mM PMSF and 0.2% BSA and homogenized using a Homogenplus Homogenizer Size S (Schuett-Biotec). The crude cell homogenate was separated using differential centrifugation steps: 400 × *g*, 10 min, 4 °C to remove cell debris, 11,000 × *g* for 10 min, 4 °C to pellet mitochondria. Mitochondria were resuspended in homogenization buffer and were used immediately or stored at −80 °C.

Cell lysates or mitochondria samples were separated on 10–18% Tris-Tricine gels and transferred onto Amersham™ Protran™ 0.2 μM nitrocellulose membranes (NC, GE Healthcare). For immunodetection, primary antibodies were incubated overnight (4 °C) as indicated (see Supplementary Table 1), secondary antibodies were incubated for 2 h at room temperature and visualized on X-ray films using enhanced chemiluminescence detection kit (GE Healthcare).

### Sucrose gradient centrifugation

Mitoplasts were purified by incubating fresh isolated mitochondria in 0.1% digitonin for 30 min on ice and 0.5 μg Proteinase K per 100 μg mitochondria for 15 min on ice. Proteinase K was blocked by addition of 2 mM PMSF followed by four washing steps. Mitoplasts (500 μg) were lysed (3% sucrose, 100 mM NH₄Cl, 15 mM MgCl₂, 20 mM Tris-HCl pH 7.5, 1% Digitonin, 1x PI-Mix, 0.08 U/μl RiboLock RNase Inhibitor) and separated by sucrose density gradient centrifugation (5–30% sucrose [w/v] in 100 mM NH₄Cl, 15 mM MgCl₂, 20 mM Tris-HCl pH 7.5, 1x PI-Mix) at 79,000 × *g* for 15 h, 4 °C using a SW41Ti rotor (Beckman Coulter). A BioComp fractionator was used to collect fractions 1–16, which were then ethanol precipitated and analyzed via western blotting.

### Blue-Native (BN) PAGE and *in gel* activity measurements

To investigate native protein complexes, mitochondria were solubilized (1% digitonin, 10 mM Tris-HCl pH 7.4, 0.1 mM EDTA, 50 mM NaCl, 10% Glycerol [v/v], 1 mM PMSF) at a concentration of 1 μg/μl for 20 min, 4 °C. Lysates were cleared from insoluble materials by centrifugation for 15 min at 21,000 × *g*, 4 °C prior to addition of BN Loading Dye (5% Coomassie Brilliant Blue G250 (w/v), 500 mM 6-aminocaproic acid, 100 mM Bis-Tris-HCl pH 7.0). Samples were separated by electrophoresis using 4–14% or 2.5–10% polyacrylamide gradient gels. Proteins were either blotted on PVDF membranes for 1 dimensional

analysis via western blotting or further separated into the 2nd dimension via 10–18% Tris-Tricine gels.

To monitor *in gel* activities, the gel was incubated either in complex I (1 mg/ml nitrotetrazoliumbluechlorid and 1 mg/ml NADH in 5 mM Tris-HCl pH 7.4) or complex IV (0.5 mg/ml diaminobenzidine, 20 μg/ml catalase, 1 μg/ml cytochrome *c* and 75 mg/ml sucrose in 50 mM KP$_i$ pH 7.4) solution[36].

### Activity measurement complex I
To determine complex I activity, the activity assay kit by abcam was used according to the manufacturer's instructions. Briefly, 200 μg of cell lysate was loaded per well and oxidation of NADH by complex I was colorimetrically detected as increase in absorbance at OD = 450 nm.

### [$^{35}$S]Methionine de novo mitochondrial protein synthesis
De novo labeling of newly synthesized mitochondrial proteins was performed as followed: Cultured cells were starved in FCS- and methionine-free media, cytosolic translation was inhibited by using 100 μg/ml emetine and incubated in the presence of 200 μCi/ml [$^{35}$S] Methionine for 1 h in fully supplemented but methionine-free DMEM media[16]. For pulse-chase labeling, cytosolic translation was inhibited using 100 μg/ml anisomycin instead of emetine and after 1 h pulse-labeling, media was changed to normal growth media and cells were harvested at indicated chase-time points. Cell lysates were subjected to SDS-PAGE followed by western blotting. Radioactive labeled mitochondrial translation products were detected using Phosphor screens and Amersham Typhoon imaging system (GE Healthcare).

### RNA isolation and northern blotting
Total RNA from cultured cells was isolated using TRIzol Reagent (Invitrogen) according to the manufacturer's instructions. RNA (2 μg) was separated on a denaturing formaldehyde/formamide 1.2% agarose gel and transferred and UV-crosslinked onto Amersham Hybond™-N membrane (GE Healthcare). RNA was visualized using [$^{32}$P]-radiolabeled probes targeting mitochondrial RNAs as indicated (see Supplementary Table 1).

### NanoString analysis
The experiment was performed following established protocols[20,37]. Briefly, equal amounts of mitochondria (100 μg) were isolated from the respective cell lines and solubilized into lysis buffer (50 mM Tris-HCl pH 7.4, 150 mM NaCl, 10% Glycerol, 10 mM MgCl$_2$, 1% Digitonin, 1 mM PMSF, 1x PI-Mix (Roche) and 0.08 U/μl RiboLock RNase inhibitor (Thermo Scientific)). RNA isolation from the lysates was performed according to the Ambion/Life Technologies protocol using TRIzol reagent and RNA Clean and Concentrator kit (Zymo Research). Isolated RNA pool was hybridized with TagSet master mix (nCounter Elements™ XT Reagents, nanoString) and probes targeting individual mitochondrial transcripts or cytosolic 18 S rRNA/5 S rRNA (Supplementary Table 2). Subsequently, the samples were analyzed in a nCounter MAX system (nanoString) following the nanoString Technologies instructions. Collected data were evaluated with nSolver software (nanoString). To assess the abundance of the transcripts of interest, the raw data were normalized to the abundance of cytosolic transcripts (18 S rRNA and 5 S rRNA).

### Quantification and statistical analysis
All experiments were carried out at least in biological triplicate and data is presented as means with standard error of the mean (SEM). Protein or RNA signals from western and northern blots were quantified using ImageJ (https://imagej.nih.gov/ij) or ImageQuant TL (GE Healthcare) and data was statistically analyzed by two-sample (equal variances) one-tailed Student's *t*-test. Statistical significance was defined by * for $p \le 0.05$, ** for $p \le 0.01$ and *** for $p \le 0.001$. Exact *p*-values are provided with the source data.

### Reporting summary
Further information on research design is available in the Nature Research Reporting Summary linked to this article.

## Data availability
Material will be available upon reasonable request and source data are provided with this paper. The original data generated in this study are provided in the supplementary information and the source data file, and are available through https://figshare.com/ with the digital object identifier https://doi.org/10.6084/m9.figshare.21276411. Source data are provided with this paper.

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

## Acknowledgements

We thank Peter Rehling for providing antibodies, Angela Boshnakovska for technical advice regarding oxygen consumption measurements and Sabine Poerschke regarding Blue-Native PAGE. This work was funded by the Deutsche Forschungsgemeinschaft: the Emmy-Noether grant [RI 2715/1-1 to R.R.-D.], the Cluster of Excellence [EXC 2067/1-390729940 to R.R.-D.] and the Collaborative Research Center 860 [SFB860/A14 to R.R.-D.]. We acknowledge support by the Open Access Publication Funds of the Goettingen University.

## Author contributions

F.N., E.L., A.K., L.D.C.-Z. and S.D. performed experiments. F.N. and R.R.-D. prepared figures. R.R.-D. designed the study and provided supervision. R.R.-D. wrote the manuscript. F.N., E.L. and R.R.-D. reviewed and edited the text.

## Funding

## Competing interests

The authors declare no competing interests.
