## [Peer Review File · Nature Communications]

Human mtRF1 terminates COX1 translation and its ablation induces mitochondrial ribosome-associated quality controlREVIEWER COMMENTS

Reviewer #1 (Remarks to the Author):

In this important, clear, and nicely delivered manuscript, Nadler et al. aim to unravel the long-standing mystery in the field of mitochondrial gene expression - the role of parallel release factors in the termination of mitochondrial translation. Despite being known for many years, the exact contribution of canonical and non-canonical mitochondrial release factors (mtRF1a and mtRF1, respectively) to the termination of protein synthesis remained uncharted. Using the cell culture model, the authors provide strong and convincing evidence explaining the function and specificity of each of these factors in mammalian mitochondria. Furthermore, they propose an interesting molecular mechanism by which deficiency in one of these factors can be overcome by the cell, emphasizing the importance of mitochondrial ribosome-associated quality control. The experiments are well designed, methodology is sound and provides a high-quality data. Undoubtedly, the work by Nadler et al. will be of interest to a broad community specialized in mitochondrial gene expression and regulation of mitochondrial function. Therefore, I strongly support the manuscript to be published in Nature Communications and have only minor comments and suggestions for the authors' consideration:

1. It is very clear that mtRF1 loss leads to isolated COX deficiency and is associated with the strong depletion of dimeric Complex IV. Yet, the Complex IV bound to high molecular weight supercomplexes (particularly to respirasome) seems fully protected. Furthermore, the mtRF1 KO cells have increased overall levels of supercomplexes in comparison to the WT cells. This is a very intriguing observation. Could it be because of the increased stability of respiratory complexes in the mtRF1 KO background? Indeed, assembly of the RC to supercomplexes has been proposed as one of the mechanisms that enhance the stability of electron transfer chain complexes. This could partially explain the phenotypes observed by the authors and add another layer to the quality control mechanism behind it. Is the half-life of the once produced MT-COX1 subunit somehow increased? Could authors speculate about it? Did authors consider performing pulse-chase labeling of mitochondrial translation products (e.g., for 24h or more) to follow MT-COX1 stability over time?

2. The isolated loss of MT-COX1 mRNA in mtRF1 depleted cells is very fascinating indeed. I fully agree with the authors on their hypothesis that mt-COX1 transcripts are produced (and possibly processed) normally, and their depletion happens due to preferential degradation. To further strengthen this hypothesis experimentally, it would be great to show that overall de novo transcription levels are not altered (e.g., by in organello transcription assay). Alternatively, the authors could block the mitochondrial translation (e.g., by chloramphenicol or doxycycline) and check if this intervention will stabilize the mt-COX1 levels in the mtRF1 KO cells. I also wondered if the authors check the levels of LRPPRC in mtRF1 cells?

3. The existence of canonical and non-canonical release factors that target a different set of respiratory complexes is very interesting from the physiological and pathological point of view as it could provide a nice mechanism to precisely deplete Complex IV from the cell (e.g. upon stress). Is something known

about the regulation of mtRF1 in stress-associated conditions such as hypoxia or in different cancer types (or cancer stages)?

4. In Figure 3.A. mtRF1a loss affects the translation of mitochondrially encoded subunits (certainly beside MT-COX1) to a different extent. For example, ND2 and ND3 seem more stabilized than ND6 or COX2/COX3, which are strongly depleted. Is this data highly replicable? If yes, what is the possible explanation for this phenomenon?

Reviewer #2 (Remarks to the Author):

Franziska Nadler et al

The question of what mtRF1, a member of the mitochondrial release factor family, does has been a mystery for a long time. Functions of the other 3 members have now been defined.

The introduction is clearly written and sets the scene for the investigation.

If an explanation is not proposed then it should still be acknowledged as present in the text.

It is not clear what is meant by “photometrically” when describing fig 2b-d. Is this just quantification of the in gel activity signals? Size markers against these gels would be helpful, as given in the B westerns in panel e.

The western for mtRF1 in panel 2f seems to show a faint signal, as does mtRF1a (as in fig 1). Is this cross reaction with another isoform of the protein that is not affected by the guide RNA KO?

Otherwise the data in figure 2 is of high quality and is consistent with the descriptions and interpretations in the text.

The metabolic labelling gels in Fig 3 are beautiful and show the difference in results obtained from siRNA depletion versus knockout lines very convincingly.

Similarly, the sucrose gradients using 5-30% show clear monosomes, which is not a common feature of many gradients. The authors have clearly established good conditions for cell harvesting, preparation and gradient parameters.

The data is of high quality throughout and is not over-interpreted.

The only issue that is not really addressed in the discussion is that the sequence of mtRF1 has insertions that have been reported as increasing the domains that would be involved in the A-site and that this would only be accommodated in the absence of mRNA in this pocket. I would ask that some space be given to discuss this point.

Methods are fine but antibody Table S1 does not indicate if the PRAB and RRDAB are made in house or gifts from collaborators as the source column is empty. If these are custom made and have not been published with then technically there should be some validation, for example a western against Rho0 and Rho+ cell lysates.

Minor points

C12ORF65 is conventionally C12orf65, which the ORF is lower case.

“While MITRAC12 is not drastically affected in both knockouts, C12ORF62 is altered.” ‘both’ should read ‘either of the’ to be more accurate.

Reviewer #3 (Remarks to the Author):

The manuscript by Nadler et al. reports the results of a comprehensive studies aiming at deciphering the molecular roles of mitochondrial release factor 1 (mtRF1), a factor the function of which has remained elusive for decades. Indeed, while the quasi-totality of mitochondrial mRNA translation is stopped thanks to conventional UAA or UAG stop-codons through mtRF1a in mammals, only two other mRNAs encoding for COX1 and ND6 possess non-conventional stops (AGA and AAG, respectively). The termination process for the latter mRNAs remained for the least poorly characterized.

In their reported work, using CRISPR-Cas9 technology, the authors generated a specific knockout cell lines where mtRF1 and mtRF1a were ablated, respectively (each in a different cell line). The authors' results show that mtRF1 is required for ensuring proper COX1 translation termination. Although ablation of mtRF1 results in isolated COX deficiency, its loss leads to the activation of mitochondrial ribosome-associated quality control pathway (mtRQC), which prevents respiratory incompetence by partial rescuing the stalled COX1-translating ribosome complexes through the degradation of COX1 mRNA.

The manuscript is globally well written, the figures are clear and the references appear to be sufficient. I warmly recommend the publication of the manuscript by Nadler et al. provided that the authors would address the few minor points below.

Minor points:

Abstract: the authors write "these results establish the role of mtRF1 in mitochondrial translation, which had been a mystery for almost 25 years,". It is preferable to remove the exact time frame (25 years), even if correct, in few years the statement will be awkward and obsolete, I would simply suggest to replace "25 years" by "decades".

In results, page 3, "Human mtRF1 and mtRF1a are required for mitochondrial function": The authors write "Loss of both mitochondrial RFs affects cellular growth as the cell number over time is significantly reduced in comparison to the wildtype control", and indeed this effect is beautifully demonstrated by figure 1c. This is not a critic per se, nevertheless I am puzzled by the limited effect of ablating mtRF1a on the cell growth (less than 2X the WT only). Maybe it's the reviewer's ignorance asking, but wouldn't it be anticipated that the total loss of mtRF1a is lethal? Perhaps it would be appropriate to explain to the non-experts how and why such mtRF1a-depleted cells can survive.

The authors do explain that premature stops were identified (in both types of ablations, mtRF1 and mtRF1a), how did the authors identify these premature stops? Could they share these data?

Related to the above, is it totally excluded that mtRF1 could at least very partially compensate for the loss of mRF1a and vice versa? The reviewer is simply trying to understand the relative limited effect of the loss of mtRF1a, perhaps the authors would care to elaborate a bit on this point.

In results page 3, “Loss of human mtRF1 leads to isolated complex IV deficiency”. The ablation of mtRF1a seems to stop totally the mitochondrial respiration (Figure 2a), it would be perhaps useful for the non-experts to briefly explain already here how does a non-respiring cell survive such a major defect.

That being said, the experiments reported in this section are simply overwhelming in quality!

Results page 4, “mtRF1 is specifically required for COX1 translation”. [35S]Met de novo labeling experiments for mtRF 1 and 1a ablated cells are beautifully executed. However, I wonder to what extent it would be possible to replicate them to test the robustness of these experiments and to add error bars to these translation rates. This is common practice when performing in vitro translation experiments (cytosolic translation). Naturally, the reviewer is aware of the extreme relative complexity of these experiments, the authors can simply comment if incapable of complying.

Same section, the authors write “As 28S small and 39S large mitoribosomal subunits as well as 55S ribosomes are detectable, defects in mitochondrial ribosome biogenesis can be excluded.”: Of course, the following is out of the scope of the current manuscript, but it is very tempting to suggest to the authors to examine the structures of their ribosomal complexes and more specifically mtRF1-terminating complexes in the mtRF1a-ablated cells, but perhaps this is already in progress...

That being said, I would suggest to tone down this strong statement based on the faith of detecting two ribosomal proteins, although the profiles are impeccable...

Reviewer #1 (Remarks to the Author)

In this important, clear, and nicely delivered manuscript, Nadler et al. aim to unravel the long-standing mystery in the field of mitochondrial gene expression - the role of parallel release factors in the termination of mitochondrial translation. Despite being known for many years, the exact contribution of canonical and non-canonical mitochondrial release factors (mtRF1a and mtRF1, respectively) to the termination of protein synthesis remained uncharted. Using the cell culture model, the authors provide strong and convincing evidence explaining the function and specificity of each of these factors in mammalian mitochondria. Furthermore, they propose an interesting molecular mechanism by which deficiency in one of these factors can be overcome by the cell, emphasizing the importance of mitochondrial ribosome-associated quality control. The experiments are well designed, methodology is sound and provides a high-quality data. Undoubtedly, the work by Nadler et al. will be of interest to a broad community specialized in mitochondrial gene expression and regulation of mitochondrial function. Therefore, I strongly support the manuscript to be published in Nature Communications and have only minor comments and suggestions for the authors' consideration:

We thank the reviewer for the positive comments on our manuscript.

1. It is very clear that mtRF1 loss leads to isolated COX deficiency and is associated with the strong depletion of dimeric Complex IV. Yet, the Complex IV bound to high molecular weight supercomplexes (particularly to respirasome) seems fully protected. Furthermore, the mtRF1 KO cells have increased overall levels of supercomplexes in comparison to the WT cells. This is a very intriguing observation. Could it be because of the increased stability of respiratory complexes in the mtRF1 KO background? Indeed, assembly of the RC to supercomplexes has been proposed as one of the mechanisms that enhance the stability of electron transfer chain complexes. This could partially explain the phenotypes observed by the authors and add another layer to the quality control mechanism behind it. Is the half-life of the once produced MT-COX1 subunit somehow increased? Could authors speculate about it? Did authors consider performing pulse-chase labeling of mitochondrial translation products (e.g., for 24h or more) to follow MT-COX1 stability over time?

We thank the reviewer for this valuable point and his/her suggestion. We performed pulse-chase labeling experiments of mtDNA-encoded proteins and followed the stability of newly synthesized COX1 for up to 24h. As shown in Supplementary Fig. S3a-b COX1 protein is indeed more stable in mtRF1^{-/-} than in wildtype control, which is in agreement with the higher stability of respiratory supercomplexes. We included and discussed these findings in the main text as indicated.

2. The isolated loss of MT-COX1 mRNA in mtRF1 depleted cells is very fascinating indeed. I fully agree with the authors on their hypothesis that mt-COX1 transcripts are produced (and possibly processed) normally, and their depletion happens due to preferential degradation. To further strengthen this hypothesis experimentally, it would be great to show that overall de novo transcription levels are not altered (e.g., by in organello transcription assay). Alternatively, the authors could block the mitochondrial translation (e.g., by chloramphenicol or doxycycline) and check if this intervention will stabilize the mt-COX1 levels in the mtRF1 KO cells. I also wondered if the authors check the levels of LRPPRC in mtRF1 cells?

We agree with the reviewer that additional experiments would strengthen our hypothesis that decreased MTCO1 levels are a consequence of mRNA degradation. As suggested we monitored the mt-RNA levels for COX1 (MTCO1) upon chloramphenicol treatment to block mitochondrial translation elongation. Under these conditions MTCO1 remains stable in mtRF1^{-/-} cells indicating that MTCO1 transcripts are degraded in a translation-dependent manner in mtRF1-deficient cells. We included these data in Fig. 4 and modified the text accordingly.

We also tested the levels of LRPPRC in *mtRF1*- and *mtRF1a*-ablated cells, but did not observe obvious changes.

3. The existence of canonical and non-canonical release factors that target a different set of respiratory complexes is very interesting from the physiological and pathological point of view as it could provide a nice mechanism to precisely deplete Complex IV from the cell (e.g. upon stress). Is something known about the regulation of *mtRF1* in stress-associated conditions such as hypoxia or in different cancer types (or cancer stages)?

To our knowledge nothing has been reported regarding the regulation of mtRF1 in the literature. We checked databases regarding the expression of mtRF1 in different cancer types, but also did not observe any interesting or conclusive findings.

4. In Figure 3.A. *mtRF1a* loss affects the translation of mitochondrially encoded subunits (certainly beside MT-COX1) to a different extent. For example, ND2 and ND3 seem more stabilized than ND6 or COX2/COX3, which are strongly depleted. Is this data highly replicable? If yes, what is the possible explanation for this phenomenon?

*We thank the reviewer for noticing this phenomenon. Yes, this data is replicable. Please, see also Fig. 3d and a further experiment below. Unfortunately, at this point we can only speculate. It might be that some stalled ribosomes are a preferable target for ribosome rescue and quality control than others, or another yet unknown alternative termination factor can compensate for the loss of *mtRF1a* to some extent. It might also reflect the stability of these proteins in their respective submodules. While ND2 and ND3 assemble at an early stage into the ND2 module, ND6 joins at an intermediate stage. We will address this phenomenon in our future studies, but it will be beyond the scope of the current study to mechanistically explain these findings.*

Reviewer #2 (Remarks to the Author)

Franziska Nadler et al

The question of what *mtRF1*, a member of the mitochondrial release factor family, does has been a mystery for a long time. Functions of the other 3 members have now been defined.

The introduction is clearly written and sets the scene for the investigation.

If an explanation is not proposed then it should still be acknowledged as present in the text.

It is not clear what is meant by “photometrically” when describing fig 2b-d. Is this just quantification of the in gel activity signals ? Size markers against these gels would be helpful, as given in the B westerns in panel e.

We apologize for being unclear. To measure complex I activity we applied two different approaches, i) in gel activity; and ii) a kit provided by abcam which allows the colorimetrically determination of NADH oxidation by complex I at OD=450 nm as described in the Methods part. We modified the text to avoid confusions.

Unfortunately, the molecular weight marker from those gels was used for western blotting as shown in e). Therefore, providing size markers would be only based on assumption, which we would like to avoid.

The western for mtRF1 in panel 2f seems to show a faint signal, as does mtRF1a (as in fig 1). Is this cross reaction with another isoform of the protein that is not affected by the guide RNA KO ?

Otherwise the data in figure 2 is of high quality and is consistent with the descriptions and interpretations in the text.

The faint signals for mtRF1 and mtRF1a in the corresponding knockouts are cross reactions of the polyclonal antibodies. We verified both knockouts with TOPO sequencing. The faint signal in mtRF1^{-/-} has a different molecular weight and the signal for mtRF1a disappears when using purified mitochondria, indicating that the mtRF1a antibody detects another non-mitochondrial protein.

The metabolic labelling gels in Fig 3 are beautiful and show the difference in results obtained from siRNA depletion versus knockout lines very convincingly.

Similarly, the sucrose gradients using 5-30% show clear monosomes, which is not a common feature of many gradients. The authors have clearly established good conditions for cell harvesting, preparation and gradient parameters.

The data is of high quality throughout and is not over-interpreted.

We thank the reviewer for this positive feedback.

The only issue that is not really addressed in the discussion is that the sequence of mtRF1 has insertions that have been reported as increasing the domains that would be involved in the A-site and that this would only be accommodated in the absence of mRNA in this pocket. I would ask that some space be given to discuss this point.

According to the convincing study by Lind et al. 2013 (Nature Communications) the insertion within the codon-recognition motifs are co-evolutionary adaptations by mtRF1 to the changes within the human mitoribosome, especially of the mt-rRNA. These authors used homology modelling and molecular dynamics simulation and show that mtRF1 has a preference for UAA and UAG codons within the A site, and not for AGA/AGG or an empty A site. We extended this point in the discussion.

Methods are fine but antibody Table S1 does not indicate if the PRAB and RRDAB are made in house or gifts from collaborators as the source column is empty. If these are custom made and have not been published with then technically there should be some validation, for example a western against Rho0 and Rho⁺ cell lysates.

The antibodies are custom made and most of them have been used in previous studies. We included references accordingly in the modified Table S1. The antibodies against ND1, CYTB and ATP6 have been validated using Rho⁰ cells as suggested by the reviewer (Fig. S6). The antibody against mtRF1 has been validated with the mtRF1^{-/-} and the corresponding rescue cell line.

Minor points

C12ORF65 is conventionally C12orf65, which the ORF is lower case.

To be consistent we used upper case when referring to the protein (C12ORF65) and lower case for DNA or mRNA (C12orf65).

“While MITRAC12 is not drastically affected in both knockouts, C12ORF62 is altered.” ‘both’ should read ‘either of the’ to be more accurate.

Text was changed accordingly.

Reviewer #3 (Remarks to the Author):

The manuscript by Nadler et al. reports the results of a comprehensive studies aiming at deciphering the molecular roles of mitochondrial release factor 1 (mtRF1), a factor the function of which has remained elusive for decades. Indeed, while the quasi-totality of mitochondrial mRNA translation is stopped thanks to conventional UAA or UAG stop-codons through mtRF1a in mammals, only two other mRNAs encoding for COX1 and ND6 possess non-conventional stops (AGA and AAG, respectively). The termination process for the latter mRNAs remained for the least poorly characterized. In their reported work, using CRISPR-Cas9 technology, the authors generated a specific knockout cell lines where mtRF1 and mtRF1a were ablated, respectively (each in a different cell line). The authors’ results show that mtRF1 is required for ensuring proper COX1 translation termination. Although ablation of mtRF1 results in isolated COX deficiency, its loss leads to the activation of mitochondrial ribosome-associated quality control pathway (mtRQC), which prevents respiratory incompetence by partial rescuing the stalled COX1-translating ribosome complexes through the degradation of COX1 mRNA.

The manuscript is globally well written, the figures are clear and the references appear to be sufficient. I warmly recommend the publication of the manuscript by Nadler et al. provided that the authors would address the few minor points below.

We thank the reviewer for his/her positive comments.

Minor points:

Abstract: the authors write “these results establish the role of mtRF1 in mitochondrial translation, which had been a mystery for almost 25 years.”. It is preferable to remove the exact time frame (25 years), even if correct, in few years the statement will be awkward and obsolete, I would simply suggest to replace “25 years” by “decades”.

We have changed the term accordingly.

In results, page 3, “Human mtRF1 and mtRF1a are required for mitochondrial function”: The authors write “Loss of both mitochondrial RFs affects cellular growth as the cell number over time is significantly reduced in comparison to the wildtype control”, and indeed this effect is beautifully demonstrated by figure 1c. This is not a critic per se, nevertheless I am puzzled by the limited effect of ablating mtRF1a on the cell growth (less than 2X the WT only). Maybe it’s the reviewer’s ignorance asking, but wouldn’t it be anticipated that the total loss of mtRF1a is lethal? Perhaps it would be appropriate to explain to the non-experts how and why such mtRF1a-depleted cells can survive.

We apologize for the missing explanation. The cells are grown in high-glucose containing media supplemented with uridine and they can survive without oxidative phosphorylation (OXPHOS) under those conditions. There are several other examples where cells deficient in mitochondrial translation can still grow/survive when cultured in high-glucose containing media. Even Rho⁰ cells, which completely lack mitochondrial DNA and are therefore OXPHOS deficient, can be cultured in high-glucose containing media supplemented with uridine. We have also used Rho⁰ cells to verify some antibodies as requested by reviewer 2 (Supplementary Fig. S6). If we would culture mtRF1a-deficient

cells in galactose-containing media to force them to produce ATP via OXPHOS, the cells would not survive. We changed the text accordingly for a better understanding.

The authors do explain that premature stops were identified (in both types of ablations, mtRF1 and mtRF1a), how did the authors identify these premature stops? Could they share these data?

We isolated genomic DNA from the cells, amplified the region targeted by the guide RNA and performed TOPO cloning and sequencing. The results are provided in Fig. S2. We extended the methods part to provide more details:

“Usage of the TOPO™-TA Cloning™ Kit (ThermoFisher Scientific) allows simple analysis of gDNA from respective clones. TOPO cloning is based on ligation of the amplified PCR product of the gDNA sequence targeted by the CRISPR guide RNA into a pCR4-TOPO TA vector. After transformation into OneShot™ competent E.coli cells, clones were selected on ampicilin LB-Agar plates. Picking a statistical relevant number of clones (≥ 20), their plasmid DNA were sequenced using M13 forward and reverse primers allowing analysis of occurring INDELS in the genome caused by CRISPR/Cas9 technology.”

Related to the above, is it totally excluded that mtRF1 could at least very partially compensate for the loss of mtRF1a and vice versa? The reviewer is simply trying to understand the relative limited effect of the loss of mtRF1a, perhaps the authors would care to elaborate a bit on this point. *Although the growth defect in high-glucose containing media is not that severe, the defect in mitochondrial function/ OXPHOS is dramatic in mtRF1a^{-/-} (Fig. 2a-e). These cells are not able to respire or to produce ATP via OXPHOS. We cannot totally exclude compensatory effects in mtRF1a^{-/-}, but this strong defect argues against a functional compensation by mtRF1. In addition, we performed siRNA-mediated depletion of mtRF1 in mtRF1a^{-/-}, but did not reveal further changes in the [35S]Methionine de novo mitochondrial protein synthesis. We also tested whether downregulation of mtRF1a would further decrease COX1 synthesis in mtRF1^{-/-}, but in contrast to C12ORF65 depletion we did not reveal any additional reduction of COX1 under these conditions.*

In results page 3, “Loss of human mtRF1 leads to isolated complex IV deficiency”. The ablation of mtRF1a seems to stop totally the mitochondrial respiration (Figure 2a), it would be perhaps useful for the non-experts to briefly explain already here how does a non-respiring cell survive such a major defect. That being said, the experiments reported in this section are simply overwhelming in quality! *We thank the reviewer for appreciating the quality of the performed experiments.*

OXPHOS-deficient cells can survive in high-glucose containing media as they obtain their energy via glycolysis. These cells acidify the media much faster as the pyruvate is converted into lactate due to the mitochondrial dysfunction, but they can still produce energy. They would not survive in galactose-containing media as mentioned above. We included a short explanation in the relevant paragraph.

Results page 4, “mtRF1 is specifically required for COX1 translation”. [35S]Met de novo labeling experiments for mtRF 1 and 1a ablated cells are beautifully executed. However, I wonder to what extent it would be possible to replicate them to test the robustness of these experiments and to add error bars to these translation rates. This is common practice when performing in vitro translation experiments (cytosolic translation). Naturally, the reviewer is aware of the extreme relative complexity of these experiments, the authors can simply comment if incapable of complying. *The graphs shown in Fig. 3a and b are just another way to visualize newly synthesized mitochondrial proteins. Due to the fact that every self-made SDS-gel runs slightly differently, every graph would have slightly shifted peaks, hence why it is not possible or extremely challenging to properly align them and to apply error bars. However, Fig. 3d-g were carried out exactly as in Fig 3a and b but here individual translation products were quantified, so that the error bars obtained here are representative.*

Same section, the authors write “As 28S small and 39S large mitoribosomal subunits as well as 55S ribosomes are detectable, defects in mitochondrial ribosome biogenesis can be excluded.”: Of course, the following is out of the scope of the current manuscript, but it is very tempting to suggest to the authors to examine the structures of their ribosomal complexes and more specifically mtRF1-terminating complexes in the mtRF1a-ablated cells, but perhaps this is already in progress... That being said, I would suggest to tone down this strong statement based on the faith of detecting two ribosomal proteins, although the profiles are impeccable...

We thank the reviewer for this suggestion and indeed we have initiated parts of this project and this work is in progress. However, as pointed out by the reviewer this part is out of the scope of this current manuscript.

We toned down the statement and modified the text as requested.

REVIEWERS' COMMENTS

Reviewer #1 (Remarks to the Author):

Many thanks to the authors for careful and extensive addressing of all my points. New experimental evidences and additional comments greatly support the initial claims of the authors and add another interesting layer to the story presented in this manuscript. Therefore, I warmly support publishing of this beautiful, important and high-quality research in Nature Communications.